# Utilizing Visual Properties to Achieve Better Representations of Objects

## Abstract

In recent years, large vision models have made significant advancements and excelled in tasks such as detection, segmentation, and tracking. This is partly due to vision models' good representation of visual objects. Although the recently proposed SAM (the Segment Anything Model Kirillov et al. (2023)) or the one/few-shot models based on SAM have wide applicability across many tasks, some researchers have found that they do not perform well on certain downstream tasks (Han et al. (2023); Tang et al. (2023)). In this paper, we focused on a specific group of these objects, which can be summarized as glass-like objects, and quantitatively studied the inadequacies related to the vision models' feature representation of glass-like objects using the representation accuracy(RA) metric we proposed. Then, we proposed a novel, extremely simple method that introduces almost no additional computations to address these inadequacies. The main idea is utilizing the visual properties of target objects to find representation dimensions which dominate in recognizing them and leveraging these information accordingly to achieve better representations of target objects. Using representation accuracy and setting these representations as reference in one-shot segmentation tasks, our experiments demonstrated the substantial effectiveness of our method.

## 1 Instruction

Research in Vision Foundation Models (VFMs) has made tremendous strides in recent times. Fueled by extensive image-text contrastive pre-training, CLIP ( Radford et al. (2021)) and ALIGN ( Jia et al. (2021)) demonstrate robust zero-shot transfer capabilities across a wide array of classification tasks. DINOv2 ( Oquab et al. (2023)) showcases remarkable proficiency in visual feature matching, enabling it to comprehend intricate information at both the image and pixel levels, solely from raw image data. Furthermore, the Segment Anything Model (SAM) ( Kirillov et al. (2023)) has achieved impressive class-agnostic segmentation performance by training on the SA-1B dataset, comprising 1 billion masks and 11 million images.

However, unlike Large Language Models (LLMs), which seamlessly integrate various language tasks using a unified model structure and pre-training approach, VFMs face challenges when directly addressing diverse perception tasks. For instance, researchers have found that SAM performs rather poorly in some downstream vision tasks, e.g. camouflaged objects( Han et al. (2023)), transparent objects( Tang et al. (2023)), mirror objects( Tang et al. (2023)), etc. It is worth emphasizing that in the testing of the transparent and mirror object segmentation task, Tang et al. manually selected the masks generated by SAM that were closest to the target objects but the performance of SAM still fell far short compared to specialized models, with nearly a 25 % difference in mean intersection over union (mIoU) compared to the previous state-of-the-art models for each task averagely. This suggests that raw features extracted by existing Vision Foundation Models may not effectively represent such objects.

To enhance the transferability of those Vision Foundation Models, efforts have been made by Per-SAM ( Zhang et al. (2023)) and Matcher ( Liu et al. (2023)) which employ a systematic approach where each object category is prompted with a single reference photo (one-shot). This process involves several key steps, including computing similarity in the patch level between the target image and reference images by features encoded by VFMs (ViT,DINOv2), extracting prompted points or

boxes to guide SAM in generating segmentation masks, and utilizing the similarity information to select segmentation masks and produce the final results.

Despite their significant advancement over previous models in traditional one/few-shot datasets, we found that they performed poorly on the one-shot glass and mirror segmentation tasks, just like SAM does. For the glass segmentation dataset, Matcher based on DINOv2 and SAM achieved an mIoU of averagely only around 40%, and for the mirror segmentation task, Mathcer's mIoU was merely around 25%, which is far below the 85% achieved on the traditional one-shot dataset FSS-1000. This indicates that existing VFMs do not represent glass surface objects as well as traditional objects. In this article, we quantitatively analyze how this representation gap manifests at the patch level, clearly showing the deficiencies of VFMs in these downstream segmentation tasks with experiments of the representation accuracy index.

After analyzing the inadequacies of existing vision models in representing glass-like objects, we propose a novel and rather simple method to improve these feature representations. The main idea is to utilize the visual properties of glass-like objects by employing a comparative approach to identify those dominant feature dimensions when recognizing these hard-to-detect objects. Specifically, because glass barriers have transparent attributes and mirrored objects have reflective qualities, we can perform comparative analysis on the same scene by adding and removing glass barriers, examining both the interior and exterior aspects of the mirrored scene. This allows us to extract the slight feature dimensions that truly characterize glass-like objects.

By processing these important dimensions, we significantly enhance accuracy and validity of the representations, which substantially improves segmentation precision in further tasks. Experimental results demonstrate the effectiveness and broad applicability of our method. In the task of glass(transparent) object segmentation, there is a average around 1% improvement in representation accuracy, with consistent improvement on the one-shot model Matcher( Liu et al. (2023)). For the mirror segmentation task, there is a consistent increase in representation accuracy, and an average improvement of around 3% on Matcher.

Our main contributions can be summarized as follows:

(i) We propose a new metric, representation accuracy to quantitatively calculated the representation accuracy of a specific vision model. We assessed the representation accuracy of several images with the encoder (DINOv2) of the recent state-of-the-art (SOTA) one-shot model across the whole datasets on the glass-like tasks, highlighting the inadequacies of existing vision models in representing glass-like objects.

(ii) We proposed a novel, clearly motivated, and easily implementable method that utilizes the visual properties of objects to extract the most important feature dimensions and process them accordingly to achieve more accurate and effective representations with almost no additional computation.

(iii) Extensive and thorough ablation experiments across three datasets demonstrate the substantial effectiveness and broad applicability of our method, providing inspiration for future research.

## 2 RELATED WORK

**Vision Foundation Models:** Driven by extensive pre-training, foundational vision models have achieved remarkable success in computer vision. Drawing inspiration from the concept of masked language modeling Devlin (2018) in natural language processing, MAE He et al. (2022) adopts an asymmetric encoder-decoder architecture and implements masked image modeling to efficiently train scalable vision Transformer models Dosovitskiy (2020). CLIP Radford et al. (2021) learns image representations from a vast corpus of 400 million image-text pairs, demonstrating impressive zero-shot image classification capabilities. Through image and patch-level discriminative self-supervised learning, DINOv2 Oquab et al. (2023) acquires versatile visual features applicable to various downstream tasks. Recently, SAM Kirillov et al. (2023), pre-trained with 1 billion masks and 11 million images, has emerged with remarkable zero-shot, class-agnostic segmentation performance. Despite the exceptional performance of vision foundation models in fine-tuning, their capabilities remain limited in various visual perception tasks.

**Vision Generalist for Segmentation:** In recent times, there has been a growing endeavor to consolidate various segmentation tasks into a unified model leveraging the Transformer architecture

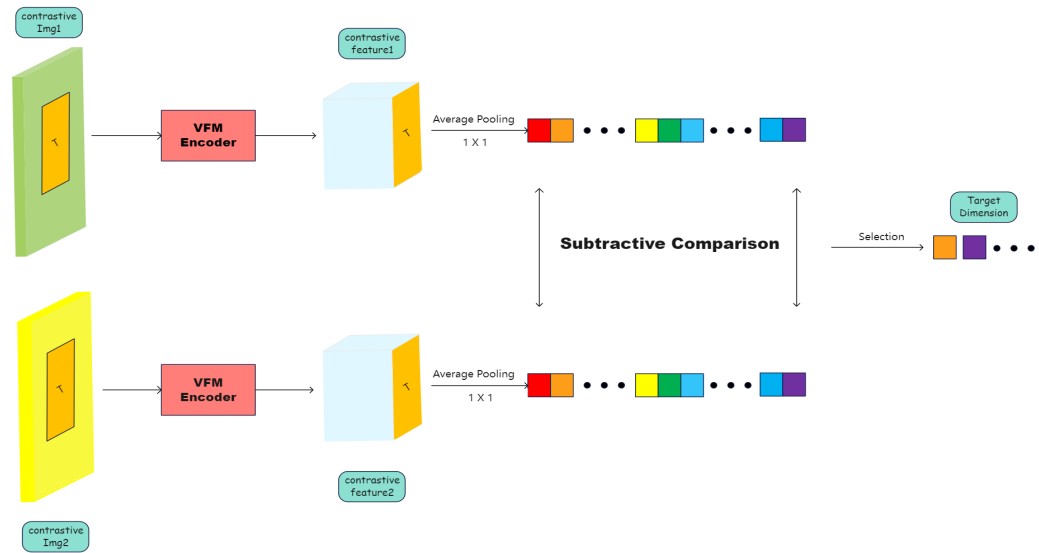

Figure 1: Illustration of our method. For a specific set of comparison images, we average-pool the features of the labeled target regions (the orange parts in the image) into one dimension. We then compute the differences to identify the dimensions with the largest changes and consistent change directions.

Vaswani (2017). The versatile Painter model Wang et al. (2023a) reimagines the outcomes of diverse visual tasks as images and employs masked image modeling on continuous pixels for in-context training with labeled datasets. SegGPT Wang et al. (2023b), a variant of the Painter model, introduces a novel random coloring method for in-context training to enhance the model's generalization capabilities. SEEM Zou et al. (2024) effectively addresses various segmentation tasks by leveraging spatial queries such as points and textual prompts. More recently, PerSAM Zhang et al. (2023) extends SAM for personalized segmentation and video object segmentation with minimal training requirements, while Matcher Liu et al. (2023), a training-free framework, endeavors to tackle various segmentation tasks in a single shot using all-purpose feature matching.

**Glass-like Object Segmentation.** Segmenting objects with a glass-like appearance presents a significantly greater challenge compared to commonly seen objects and this heightened difficulty arises primarily from the fact that the inner regions of glass objects often exhibit a perplexing similarity to their surrounding backgrounds. To address this issue, some methods Mei et al. (2022) have turned to the utilization of additional multi-modal information, such as 4D light-field data, refractive flow maps, thermal imaging and spectral polarization. Recent contributions from researchers such as Lin et al. (2021; 2020), have led to the creation of large-scale RGB image datasets specifically tailored to glass-like objects, promoting this research in the community. Additionally, Lin et al. (2021) have introduced methods for the segmentation of glass-like objects with the aid of boundary cues, leveraging the precise localization afforded by boundaries. After the emergence of SAM, Tang et al. (2023) tested its performance on such tasks. They manually selected the highest-scoring masks that matched the ground truth to assess SAM's capability in this task, and found that SAM did not perform well as other tasks. Despite the manual selection, SAM still lagged far behind the aforementioned specialized models.

## 3 PROBLEM ANALYSIS

It has become a fundamental consensus that similar images or similar image patches encoded by a pre-trained vision model produce similar features. This is specifically reflected in the dimensions of their respective features, where similar features display values that have the same sign and are closely sized in specific dimensions, allowing for the use of metrics such as cosine similarity to represent the similarity between two features. Recently Liu et al. (2023) have leveraged this similarity to propose a model called Matcher, which generates prompt points or prompt boxes by comparing

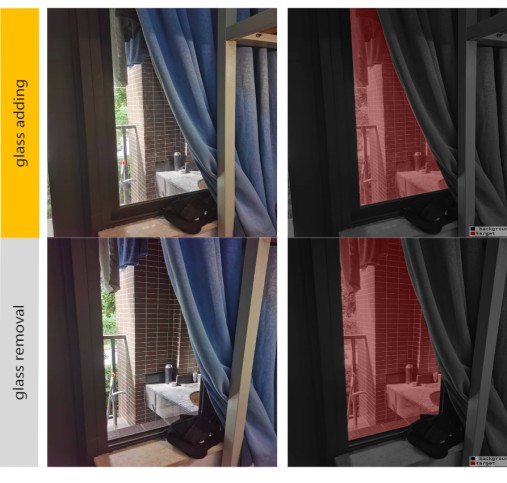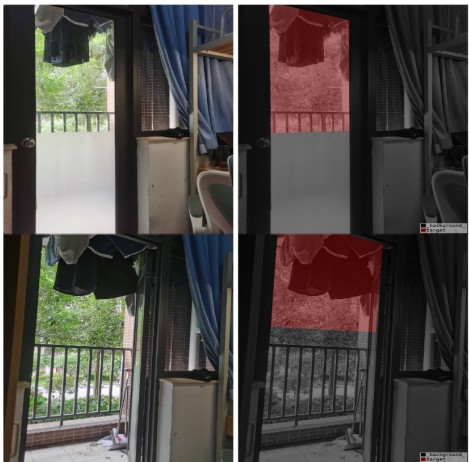

Figure 2: An example of comparative images regarding the addition and removal of glass barriers, with the target regions labeled

the similarity between reference image patches and target image patches to create and select masks produced by SAM. Matcher uses DINOv2 with a ViT-L/14 as the default image encoder and also in this paper authors found that DINOv2 has better patch-level representation ability than SAM, which promotes exact patch matching between different images so it can be considered that DINOv2 is the best VFM for representing similarities at the patch level. Matcher has significantly outperformed previous models in traditional one-shot tasks, such as COCO-$20^i$, FSS-1000.

We followed this approach and tested the Matcher model on the glass-like object task, which specifically includes two major categories: glass object segmentation and mirror segmentation. We found that Matcher performed poorly on this task. Regarding this result, we first considered potential issues in the similarity matching process due to the inadequate representation of glass-like objects. To verify our idea, we conducted the following experiment:

We defined a simple and intuitive metric, representation accuracy(RA). This metric calculates the probability that (the most similar patch in the target image) of (patches belonging to the reference object in the reference image) are belonging to the (target object in the target image). The similarity is evaluated by the cosine similarity, shown as Formula 1. The representation accuracy(RA) can be mathematically represented by Formula 2 and the final result of the RA metric is the average of the RA values for each image in the entire dataset based on the selected reference image. We randomly selected several images from each dataset as reference images to test the representation accuracy across the entire dataset, and the results are shown in Table 2. From the experimental results, it is evident that the average representation accuracy is poor, with a substantial number of patches that do not belong to the target object being incorrectly matched. This indicates a deficiency in vision models' representation of objects made of glass materials.

$$S = \frac{\boldsymbol{f}_r \cdot \boldsymbol{f}_t}{||\boldsymbol{f}_r|| \cdot ||\boldsymbol{f}_t||} \quad (1)$$

Where $f_r$ indicates the feature of a patch or a pixel of the reference image and $f_t$ is the feature of a patch or a pixel of the target image.

$$\mathbf{P}(\mathbf{p}_t^i \in M_t \mid \mathbf{p}_r^i \in M_r) \quad (2)$$

Where $\mathbf{p}_r^i$ represents the i-th patch belonging to the mask of reference image ($M_r$) and $\mathbf{p}_t^i$ represents the matched patch in the target image of $\mathbf{p}_r^i$. $M_r$ Represents the mask of the target image.

Glass-like objects, whether glass barriers or mirrors, differ from other objects (animals with various parts) in that their individual components do not exhibit significant differences. Thus, we can select any image and calculate the RA of the reference part of the image. At the image level, we can test

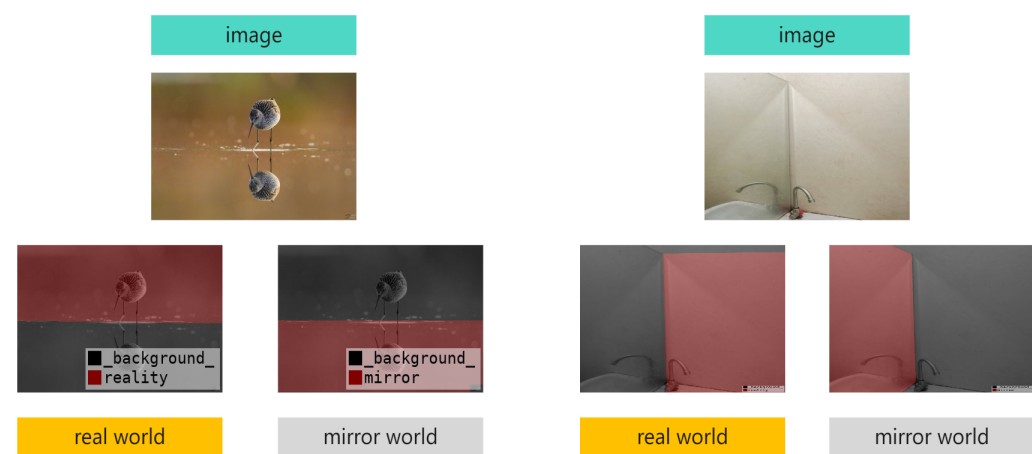

Figure 3: An example of comparative images regarding the interior and exterior aspects of the mirrored scene, with the target regions labeled

the ability of a specific image to serve as a reference, while at the level of multiple image groups, we can evaluate the overall representation capability of a vision model for that object. The RA metric can test the representation ability of vision models without relying on specific tasks (such as one-shot segmentation or classification) and can serve as a foundational metric for various tasks and datasets that rely on similarity computing.

We analyzed features from several images on respective dataset using the RA metric and the results are shown as Table 1. We selected a few images from each dataset as references and tested their RA metrics on their respective datasets, as shown in Table 1. The experimental results indicate that for the PMD dataset, the average RA metric is only about 42%; for the GSD dataset, the average RA metric is approximately 60%; and for the Trans10k dataset, the average RA metric is around 50%. This means that averagely in every experimental case, nearly half points were incorrectly labeled, causing significant interference for mask generators like SAM. This is primarily due to the significant interference caused by the transmitted or reflected backgrounds in glass objects. The vision model inevitably introduced a large amount of irrelevant background information during the encoding process of single images, which negatively impacted the representation of glass objects.

## 4 METHODOLOGY

As mentioned in the problem analysis section, glass-like objects almost always have a complex background mixed within them, resulting in many dimensions of the features encoded by the vision model being used to store this background information. This leads to numerous errors in patch matching when calculating similarity using these features. To address this issue, we propose a novel and targeted approach.

Since the transparency and reflectivity of glass-like objects interfere with the vision model, we can leverage these two visual properties to design a series of comparative scenarios. For example, we can add and remove glass barriers (e.g., windows) in the same location or examine the inside and outside of mirrors in the same place, seeking the dimensions where the average feature values change the most with consistent direction. These dimensions may, to some extent, better represent glass-like objects. Subsequently, we can further process these dimensions for either the reference images or the target images, such as feature dimensions reduction, to enhance the accuracy of the matching process.

Specifically, we set up five pairs of images comparing the addition and removal of glass barriers, along with six pairs of images contrasting the inside and outside of mirrors, labeling target regions within these images. We then averaged features of patches in target regions of each image and calculated dimensions with the largest absolute value changes for each pair of features. By identifying

Table 1: Quantitative RA results of selected images on three datasets. Note that '+' means improving Matcher by our dimension-based method.

| Methods | GSD (glass dataset) | | | |
| | Img1 | Img2 | Img3 | - |
|---|---|---|---|---|
| Matcher | 52.17 | 70.67 | 63.44 | - |
| Matcher+ | 52.74 | 71.63 | 64.34 | - |
| | PMD (mirror dataset) | | | |
| | Img1 | Img2 | Img3 | Img4 |
| Matcher | 30.26 | 48.93 | 48.09 | 41.07 |
| Matcher+ | 30.42 | 49.22 | 48.40 | 42.79 |
| | MSD (mirror dataset) | | | |
| | Img1 | Img2 | Img3 | - |
| Matcher | 59.65 | 40.32 | 32.31 | - |
| Matcher+ | 60.10 | 40.80 | 32.90 | - |

several dimensions that appeared most frequently with consistent change directions, we established these as target dimensions. In practice, we identified seven dimensions that exhibited the greatest changes with consistent directions. These seven dimensions consistently increased or decreased with the introduction of glass, indicating that the values of these dimensions for glass-like objects should generally be positive or negative. These dimensions capture the most information about glass-like objects.The selection process can be mathematically represented as follows:

$$\mathbb{S}_i = \{argmax(|\mathbf{F}_i - \overline{\mathbf{F}_i}|,\ n)\} \tag{3}$$

$$\mathbb{I} = \{most(\mathbb{S}_1 \cup \mathbb{S}_2 \cup \cdots \mathbb{S}_i)\} \tag{4}$$

$\mathbf{F}_i$ and $\overline{\mathbf{F}_i}$ represent a pair of features for the comparative target regions, where n indicates the number of dimensions for selecting the top n values. $\mathbb{S}_i$ is the dimension set selected from the i-th pair of comparison images, and the $most$ function identifies the dimensions that appears most frequently and has a consistent direction of change among all these selected dimensions. The detailed selection process and the chosen dimensions can be found in the appendix.

After obtaining these important dimensions, we adopted a very simple operational approach. We believe that if our idea is correct, then based on the direction of change of the corresponding dimensions we identified, we can achieve significant improvements in the model by simply adjusting the corresponding dimensions of reference features of reference images using simple slight addition and subtraction. Compared to directly changing the sign, this method can be effective on every feature while introducing almost no additional computation, and it helps prevent excessive changes to the overall numerical distribution of the features, which could lead to the loss of information from many other dimensions. For a specific task, the dimensions we adjust and the values we modify remain constant, and the adjustment values for each dimension are the same. Note that our adjustments are only applied to the reference part, and this process can be mathematically represented as:

$$\mathbf{F}^o = \mathbf{F} \cdot M_r \tag{5}$$

$$\mathbf{F}^o_{:,i} = \mathbf{F}^o_{:,i} \pm \lambda,\ if\ i\ in\ \mathbb{I} \tag{6}$$

$$\mathbf{F}^o = \mathbf{F}^o + \mathbf{F} \cdot (1 - M_r) \tag{7}$$

$\mathbf{F}$ represents the original features of the reference image, including the reference part and non-reference part. The equation (5) retains only features of the reference patches of the reference image, meaning that the adjustment operation only affects features of the reference patches. The equation (6) adds dimensions of all patches that belong to the reference part by $\lambda$, with the sign determined

Table 2: Quantitative results of selected images on PMD dataset. Note that '+' means improving Matcher by our dimension-based method.

| PMD | | | | | |
|---|---|---|---|---|---|
| Methods | Backbone | Img1 mIoU | Img2 mIoU | Img3 mIoU | Img4 mIoU |
| Matcher | DINOv2/SAM | 28.41 | 25.92 | 12.67 | 24.93 |
| Matcher+ | DINOv2/SAM | 31.34 | 26.52 | 15.59 | 26.19 |

by the change directions we identified in the selected dimension set. The equation (7) places values of non-reference patches back into the output features $\mathbf{F}^o$.

In practice, these three operations can be completed with minimal code and very few computations. Experiments have demonstrated the significant effectiveness of our rather simple operation.

## 5 EXPERIMENTS

### 5.1 DATASETS

We evaluate the proposed method on three widely used glass and mirror datasets i.e., MSD Yang et al. (2019), GSD Lin et al. (2021) and PMD Lin et al. (2020). MSD is a large mirror segmentation dataset with 4018 images in total and 955 images for test. GSD is a medium-scale glass segmentation dataset containing 4,098 glass images, covering a diversity of indoor and outdoor scenes. All the data are randomly split into a training set with 3,285 images and a test set with 813 images. PMD is another large-scale mirror dataset containing 5,096 training images and 571 test images. It has a variety of real-world images that cover diverse scenes and common objects, making it much closer to practical applications.

### 5.2 EVALUATION METRICS

mIoU (mean Intersection over Union) is applied in our experiments, which is a common evaluation metric used in image segmentation tasks, particularly for semantic segmentation and instance segmentation.

Intersection over Union (IoU) measures the overlap between the predicted segmentation and the ground truth segmentation for a particular class, which is calculated as follows:

$$IoU = \frac{Area\ of\ Overlap}{Area\ of\ Union} = \frac{|A \cap B|}{|A \cup B|} \tag{8}$$

Where A is the predicted segmentation mask and B is the ground truth segmentation mask.

### 5.3 EXPERIMENTAL SETTING

During the experiment, we randomly selected several images from three datasets respectively for one-shot reference and tested the performance of the original model Matcher against the model improved using our method across each dataset. The settings for all parameters of the Matcher can be found in the appendix.

For the MSD dataset, we randomly selected three images from its training set; for the GSD dataset, we randomly selected three images from its test set; and for the PMD dataset, we randomly selected four images from the training set. The names of all the images can be found in the appendix. For the two mirror segmentation datasets, the constant $\lambda$ was set to 6, while for the glass segmentation dataset, the constant $\lambda$ was set to 4. Note that the results for the PMD dataset represent the average test results of six separate sub-datasets.

Table 3: Quantitative results of selected images on GSD dataset. Note that '+' means improving Matcher by our dimension-based method.

| GSD | | | | |
|---|---|---|---|---|
| Methods | Backbone | Img1 mIoU | Img2 mIoU | Img3 mIoU |
| Matcher | DINOv2/SAM | 48.35 | 41.35 | 45.92 |
| Matcher+ | DINOv2/SAM | 48.48 | 41.66 | 46.51 |

Table 4: Quantitative results of selected images on MSD dataset. Note that '+' means improving Matcher by our dimension-based method.

| MSD | | | | |
|---|---|---|---|---|
| Methods | Backbone | Img1 mIoU | Img2 mIoU | Img3 mIoU |
| Matcher | DINOv2/SAM | 17.93 | 25.71 | 33.42 |
| Matcher+ | DINOv2/SAM | 18.20 | 25.80 | 33.83 |

## 5.4 RESULTS

We compare the original model Matcher and the improved model using our method (Matcher+) on GSD, PMD, and MSD dataset. Note that, apart from processing dimensions of reference features based on the information obtained from the comparative scenes of glass and mirror objects, there are no changes in the other parts. In other words, both models have the exact same parameter settings.

As illustrated in Table 1, Matcher+ achieves average RA of 62.90%, 42.71%, 44.6% on GSD, PMD, MSD respectively, with 0.81%, 0.62%, 0.51% enhancement over the original model. It is worth emphasizing that all the images showed significant improvements, demonstrating the validity and applicability of our method for achieving better representations of glass-like objects. The RA metric is not dependent on any specific task and can fundamentally reflect the improvement in representational capabilities.

As seen in Table 2,3,4, Matcher+ can achieve average mIoU of 24.91%, 45.55%, 25.94% on PMD, GSD, MSD respectively, with 1.93%, 0.34%, 0.25% enhancement over the original model. All the images exhibited significant improvements after getting better representations as reference. These experimental results justify the effectiveness and broad applicability of our method, as it can bring significant improvements to the model with minimal operations.

## 6 CONCLUSION

In this paper, we first tested the accuracy of existing vision foundation models in representing glass-like objects. Specifically, we introduced a metric called representation accuracy and pinpointed significant inadequacies in current glass-like object representations after testing these representations applied to a one-shot segmentation model. We analyzed the reasons for these deficiencies and concluded that background information within the glass severely interfered with vision models' representations of glass-like objects. To address this issue, we leveraged the visual properties of glass-like objects, designing several sets of contrastive scenarios to identify dimensions that best represent glass-like objects, and applied a simple method to these dimensions with almost no additional computational overhead. Remarkably, using only eleven pairs of contrastive scenarios, getting only seven dimensions and very simple operations resulted in a significant improvement in model's performance, proving the correctness of our ideas and the substantial effectiveness of our approach.

**Limitations and Future Research**.The number of contrastive scenarios we designed was limited and rather straightforward, and or processing method was simple with minimal additional computation. We believe that, moving forward, the design of more diverse and rich contrastive scenarios,

along with improved dimensional processing methods such as dimension reduction, could enhance the information obtained through comparison and be effective across different vision models. Furthermore, we believe that the method of leveraging visual characteristics to obtain representations can be applied to various tasks, providing significant inspiration to researchers in other fields.

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

# A APPENDIX

## A.1 MATCHER SETTING

All parameters in the matcher remained unchanged during the experiment, including in the subsequent improved models. Specifically, $\alpha$, $\beta$, and $\lambda$ were set to 0.8, 0.2, and 1.0, respectively. The number of masks was set to 2, and the sample range was set to (1, 3).

## A.2 SELECTING PROCESS AND FINAL RESULTS

Encoder : DINOv2

Channel Total : 1024

A: the averagely pooled reference of glass or mirror

B: the averagely pooled contrastive feature

*Glass Objects*

**pair1:**

The 10 channels with the largest increases {706,471,865,83,250,353,439,564,860,664}

Channel 706: A - B = 0.5744844079017639

Channel 471: A - B = 0.5549602508544922

Channel 865: A - B = 0.48515594005584717

Channel 83: A - B = 0.4838593006134033

Channel 250: A - B = 0.4813923239707947

Channel 353: A - B = 0.479403018951416

Channel 439: A - B = 0.4780835509300232

Channel 564: A - B = 0.461056113243103

Channel 860: A - B = 0.44586917757987976

Channel 664: A - B = 0.4371333420276642

the 10 channels with the largest decreases {52,746,926,351,210,262,947,878,279,557}

Channel 52: B - A = 0.622995138168335

Channel 746: B - A = 0.5467661619186401

Channel 926: B - A = 0.5078479647636414

Channel 351: B - A = 0.5003160834312439

Channel 210: B - A = 0.46753764152526855

Channel 262: B - A = 0.44749927520751953

Channel 947: B - A = 0.42647767066955566

Channel 878: B - A = 0.4223964214324951

Channel 279: B - A = 0.4098420739173889

Channel 557: B - A = 0.4038591682910919

**pair2:**

The 10 channels with the largest increases {353,471,30,901,249,535,896,436,142,457}

Channel 353: A - B = 2.8375134468078613

Channel 471: A - B = 1.175917387008667

Channel 30: A - B = 0.7881667017936707

Channel 901: A - B = 0.558587908744812

Channel 249: A - B = 0.5483311414718628

Channel 535: A - B = 0.5433468222618103

Channel 896: A - B = 0.5429238677024841

Channel 436: A - B = 0.530151903629303

Channel 142: A - B = 0.5292552709579468

Channel 457: A - B = 0.5265811681747437

the 10 channels with the largest decreases {746,230,52,491,842,57,279,103,737,865}

Channel 746: B - A = 1.1945500373840332

Channel 230: B - A = 0.8458571434020996

Channel 52: B - A = 0.8162980079650879

Channel 491: B - A = 0.723149299621582

Channel 842: B - A = 0.7093675136566162

Channel 57: B - A = 0.6662235260009766

Channel 279: B - A = 0.5874816179275513

Channel 103: B - A = 0.569592297077179

Channel 737: B - A = 0.5546806454658508

Channel 865: B - A = 0.537898063659668

**pair3:**...

**pair4:**...

**pair5:**...

**Final**:

The dimensions that appear most frequently and have a consistent direction of change {30(+),230(-),353(+),746(-),947(-)}.

The remaining data is available in supplementary material and all comparison images will be publicly available after the article is accepted.

*Mirror Objects*

 **pair1:**

The 10 channels with the largest increases {663,103,372,720,751,530,293,811,933,354}

Channel 663: A - B = 1.3494985103607178

Channel 103: A - B = 1.325232744216919

Channel 372: A - B = 1.2999900579452515

Channel 720: A - B = 1.2474082708358765

Channel 751: A - B = 1.2384324073791504

Channel 530: A - B = 1.2309551239013672

Channel 293: A - B = 1.2151052951812744

Channel 811: A - B = 1.1934257745742798

Channel 933: A - B = 1.1849133968353271

Channel 354: A - B = 1.1797757148742676

the 10 channels with the largest decreases {42,160,588,646,436,352,916,950,292,121}

Channel 42: B - A = 1.269151210784912

Channel 160: B - A = 1.254387617111206

Channel 588: B - A = 1.2217018604278564

Channel 646: B - A = 1.212447166442871

Channel 436: B - A = 1.20435571670532233

Channel 352: B - A = 1.2021822929382324

Channel 916: B - A = 1.1839725971221924

Channel 950: B - A = 1.183129072189331

Channel 292: B - A = 1.1283351182937622

Channel 121: B - A = 1.1268000602722168

**pair2:**

The 10 channels with the largest increases {609,133,962,199,649,480,629,546,415,641}

Channel 609: A - B = 1.7402980327606201

Channel 133: A - B = 1.5993781089782715

Channel 962: A - B = 1.5648748874664307

Channel 199: A - B = 1.5560945272445679

Channel 649: A - B = 1.5514092445373535

Channel 480: A - B = 1.5371569395065308

Channel 629: A - B = 1.5332424640655518

Channel 546: A - B = 1.5257086753845215

Channel 415: A - B = 1.5213422775268555

Channel 641: A - B = 1.5193612575531006

the 10 channels with the largest decreases {664,455,367,29,384,755,957,1021,724,122}

Channel 664: B - A = 1.7092516422271729

Channel 455: B - A = 1.680405855178833

Channel 367: B - A = 1.6586047410964966

Channel 29: B - A = 1.628407597541809

Channel 384: B - A = 1.624773621559143

Channel 755: B - A = 1.516843557357788

Channel 957: B - A = 1.487288236618042

Channel 1021: B - A = 1.4855324029922485

Channel 724: B - A = 1.4777144193649292

Channel 122: B - A = 1.4625048637390137

**pair3:**...

**pair4:**...

**pair5:**...

**pair6:**...

**Final:**

The dimensions that appear most frequently and have a consistent direction of change $\{384(+), 663(-)\}$.

The remaining data is available in supplementary material and all comparison images will be publicly available after the article is accepted.

A.3 IMAGE ID

**PMD**

Img1:1269.jpg,

Img2:000000251062.jpg

Img3:43.jpg

Img4:150.jpg

**GSD**

Img1:2019_11_07_14_11_IMG_1267.jpg

Img2:7438807072_042d612827_k.jpg

Img3:11477025646_c856bdf3d6_b.jpg

**MSD**

Img1:1_512x640.jpg

Img2:2_512x640.jpg

Img3:5_512x640.jpg

