# OpenReview forum: "Utilizing Visual Properties to Achieve Better Representations of Objects"
_ICLR.cc/2025/Conference — Submitted to ICLR 2025_

### Official Review · Reviewer_3ufn · 2024-10-22

**Soundness:** 1
**Presentation:** 1
**Contribution:** 1
**Rating:** 1
**Confidence:** 4

**Summary:**

The paper introduces a novel metric to assess the effectiveness of Vision Foundation Models (VFMs) in representing and segmenting glass-like objects. The authors evaluate the performance of Matcher [1] on this type of data, and conclude that VFMs struggle to accurately represent glass-like objects. To address this limitation, the paper proposes an alignment method to enhance the representations of glass-like objects specifically for Matcher-based downstream tasks. This approach is tested on three distinct datasets, showing some improvements in segmentation performance.

**Strengths:**

- The proposed method is computationally efficient (at "training" time) and requires minimal learnable parameters or hyperparameter tuning.
- The comparative study using real-world pairs of images with and without the glass-like objects is original.

**Weaknesses:**

- The overall clarity of the paper needs improvement. I recommend starting with a clear introduction to the problem and a stronger motivation for why it is important to address.
- The writing throughout the paper is often difficult to follow, affecting readability.
- The proposed method is based on a comparative study using a very limited number of image pairs. It is unclear how the conclusions drawn from such a small sample size can be generalized. This issue is evident, for example, in the dataset dependence of the parameter $\lambda$. The method also seems to overlap with what could be achieved by training an adapter on top of DINOv2 [2] features using the reference images.
- The method relies on large Vision Foundation Models (VFMs) such as DINOv2 [2] and SAM [3]. Comparing its performance to standard segmentation approaches, such as a linear segmentation head on top of frozen features, would be useful to justify the high computational cost of the proposed method at inference and in general to put things into perspective.
- The overall performance improvements are modest, raising questions about the method’s practical impact.

**Questions:**

- Can you clarify the distinction between the proposed metric (representation accuracy) and the accuracy achieved by a $k$-NN classifier at the patch level (see Hummingbird [4])? What additional insights does the proposed metric offer?
- Why are the results in the tables not reported as averages over the entire datasets?

**References**

[1] Matcher: Segment Anything with One Shot Using All-Purpose Feature Matching. The Twelfth International Conference on Learning Representations (ICLR), 2024.

[2] DINOv2: Learning Robust Visual Features without Supervision. Transactions on Machine Learning Research (TMLR).

[3] Segment Anything. Proceedings of the IEEE/CVF International Conference on Computer Vision (ICCV), 2023.

[4] Towards In-Context Scene Understanding. Advances in Neural Information Processing Systems (NeurIPS), 2024.

---

### Official Review · Reviewer_KPFD · 2024-10-22

**Soundness:** 1
**Presentation:** 1
**Contribution:** 1
**Rating:** 1
**Confidence:** 5

**Summary:**

This paper presents a feature modification approach for DINOv2 features aimed at enhancing Matcher’s [1] ability to segment glass-like objects. To achieve this, authors identified the feature directions corresponding to glass-like appearances by comparing features from glass or mirror regions using a manually labeled subset of 11 image pairs. The approach was evaluated on a subset of either 3 or 4 images from different datasets.

---
[1] Yang Liu et al., Matcher: Segment anything with one shot using all-purpose feature matching., arXiv

**Strengths:**

* The high-level motivation of modifying foundation model features for specific object types is sound.

**Weaknesses:**

The paper is difficult to follow, lacks a logical flow in its presentation, and is missing a convincing motivation and generalizable results. The notation is improperly used and inconsistent throughout. Additionally, the paper does not adhere to general writing guidelines, and figures are not referenced in the text.

**Structural Problems**
* The concept of Representation Accuracy (RA) is discussed at length, yet it lacks a formal definition or a clear explanation in plain English. RA is neither well-motivated nor supported by any explanation or experiment validating its usefulness as a metric. Furthermore, Equation 2 is not adequately explained as presented.
* All experiments are conducted on randomly sampled image sets of only 3 or 4 images, which is insufficient to support claims of generalization. As a result, the claims in the paper are limited to being a proof of concept for manually editing features for a small number of images, rendering all algorithmic claims unsupported.
* In Table 1, due to the lack of motivation and explanation for RA, the experimental results and conclusions appear disconnected. Since the experiments are based on just 3 or 4 selected images, the results presented in Table 2 are also not valid.


**Some Writing Issues**
* Section 3 contains several notation issues. For instance, in Equation 1, the term $S$, which is not mentioned in the text, should have a subscript, $S_{rt}$, as it is defined over patches $r$ and $t$. Line 198 refers to Table 2 for RA comparison, but Table 2 presents an mIoU comparison. In line 212, there are misused variables, with “target image of $p^i_r$” actually referring to “target image of $M_t$” and “$M_r$” should be “$M_t$” in the sentence “Represents the mask of the target image,” based on context.
* The usage of variables in Equation 3 and the corresponding text is inconsistent.
* In line 302, it says "$F_i$  and  $F_i$" , but these should be two different terms.

**Questions:**

Please see the weaknesses.

---

### Official Review · Reviewer_SQan · 2024-10-29

**Soundness:** 2
**Presentation:** 1
**Contribution:** 1
**Rating:** 3
**Confidence:** 3

**Summary:**

The paper is about segmenting reflective objects. First, they show that local patch representations of glass/reflective surfaces mostly captures the underlying background (and the the glass object itself). Then they propose a feature engineering process to alleviate this effect. They show slight improvements on NN patch retrieval tasks.

**Strengths:**

-The paper sheds light on the not so common problem of segmenting reflective objects.

**Weaknesses:**

-The manuscript is contains many spelling/grammar mistakes e.g. L081.

-The citations are not properly handled (citep vs citet) e.g. L097.

-Some claims are too bold and not justified e.g. “Matcher uses DINOv2 with a ViT-L/14 as the default image encoder and also in this paper authors found that DINOv2 has better patch-level representation ability than SAM, which promotes exact patch matching between different images so it can be considered that DINOv2 is the best VFM for representing similarities at the patch level.”

-The RA metric is not novel, it is the accuracy of a NN retrieval classifier.

-The paper hard to read. The introduced notation does not dislose scalar/matrix/tensor dimensions which makes it confusing.

-The addition and removal of glass barriers is not clearly defined. The definition of a “glass barrier” is also unclear to me at this point. Figure 2 supposedly explains this but there is not pointer to that figure in the text if I am not mistaken.

-From a high-level point of view, what the authors are doing is labeling additional data. I don't think their method is superior to training/finetuning the feature extractors with the additional labeled data.

**Questions:**

-Overall, I don't think a rebuttal would clear doubts I have and think the authors should deeply revise the paper. The content could be improved by adding additional data-driven baselines (i.e. machine learning based) and by taking into consideration the suggestions listed in the weaknesses section.

---

### Official Review · Reviewer_HqNi · 2024-11-01

**Soundness:** 1
**Presentation:** 1
**Contribution:** 1
**Rating:** 3
**Confidence:** 4

**Summary:**

Although current VLMs provide feature representations that can be adapted to downstream tasks, these features are less effective on glass-like object segmentation task. This paper proposes a new metric called representation accuracy and a simple method for segmenting glass-like objects. Specifically, the main idea is to utilize the visual properties of target objects to find representation dimensions which dominate in recognizing them. Given such information, specific representations are extracted regarding these target objects.

**Strengths:**

1) A new metric called representation accuracy is defined to compute the representation accuracy of a specific vision model. This metric is used to test with DINOv2 on glass-like datasets, showing that current VLMs are less effective in segmenting these glass-like objects.

2) A new method is proposed to utilize the visual properties of objects to extract the most important feature dimensions to achieve better representations. It takes no extra computation or any other training.

3) The experiments are conducted on three datasets showing the efficacy of the proposed method on glass-like segmentation task.

**Weaknesses:**

1) Regarding the definition of representation accuracy in Equation 2, what is the definition mask of the target (M_t) and reference image (M_r)? The paper lacks of detailed explanation of defining the masks and the way to compute them.

2) In methodology, how to find the image pairs (comparative images) that show semantic similarity? Do you define the comparative images as semantically similar with slight visual differences?

3) What is the definition of subtractive comparison among the features? Any math equation referring to it?

4) In the captions of Figure 3, is it possible to provide further explanation on the "interior and exterior aspects of the mirrored scenes"?

**Questions:**

Please refer to the questions listed in "Weaknesses".

---

### Official Review · Reviewer_8qBQ · 2024-11-02

**Soundness:** 2
**Presentation:** 1
**Contribution:** 1
**Rating:** 3
**Confidence:** 4

**Summary:**

The paper considers the difficult problem of dealing with glass and mirror objects in images.
It demonstrates, based on a newly introduced metric, the poor performance of representations from standard vision foundation models, in particular in the form of the one-shot segmentation method Matcher, when evaluated specifically on such data. The paper then proposes a scheme to select, in a supervised manner based on a few images, feature dimensions that are most affected by the addition/removal of the glass. By 'correcting' these dimensions before applying the Matcher algorithm, they show a small improvement in the results on three datasets.

**Strengths:**

- The paper tackles a challenging open problem in visual understanding.
- A new metric is introduced.
- The core idea of selecting feature dimensions related to the effect of glass or mirrors has some potential, but should be worked out in a more clever way.

**Weaknesses:**

- The paper is badly structured and, as a consequence, hard to follow. Figures are not referred to in the text. Text has wrong references to tables (e.g. l. 199 refers to Table 2, that has mIOU results while text discusses RA results). There are a lot of forward references, e.g. section3 on problem analysis discusses results of tables from section 5 on experimental results, without telling the reader what data is used or what the exact setup is.

- Overall, the above point makes it hard to know precisely what the authors did exactly. I had to make some guesses at several points. Most importantly, I'm still not sure about the actual task they are performing / evaluating. In some parts it's suggested this is about  segmentation of glass/mirror objects. But in other places it's about matching between a reference image and a target image.  I assume what the authors did in the end is close to the one-shot segmentation of Matcher.

- The method is not described rigorously, making it impossible to reproduce the results. In particular, the 'most' function in eq. 4 is only vaguely described. The text refers to the appendix for more details, but their only numerical values are given, still not explaining the precise algorithm.

- The reported results are anecdotal. Results are reported only for 3-4 images per dataset. It's unclear which images these are and how they were selected.  At the very least, averaged results over the entire dataset should be reported.

- Results show only a minor improvement (in the range of 1 or 2 %) over the very poor results of the baseline. Results are reported for one set of (manually selected?) training images. No details on how these images were selected are given. At the very least, the results should have been repeated for different sets of training images, so the standard deviation on the results could be added to the tables and the reader could get an idea on whether these results are significant or not.

- Given that a new metric is used, more naive baselines should be added: what RA values would one get with a random representation ? Are the numbers reported significantly better ?

- The whole paper builds on one baseline work, Matcher. The proposed method is applied only on top of that method and the results are only compared against that method.  Other state-of-the-art methods, or extra baselines, should be added to the comparison. There is no further analysis, such as a sensitivity analysis of the hyperparameters used, an ablation study or comparison of different variants of the method (e.g. determining the lambda parameter for each of the selected dimensions separately, based on the observed differences in the training data). There are no qualitative results included neither.

**Questions:**

There are a lot of improvements necessary to bring this paper to the level required by ICLR.
I have several questions, mostly related to clarifying confusion (see above), but I don't think any answer will make me change my opinion on this paper, as it's lacking in several directions (contribution, clarity, experimental validation).

---

### Meta-Review · Area_Chair_n2Kh · 2024-12-19

**Metareview:**

Summary:
This paper proposes a feature modification approach for DINOv2 features aimed at enhancing Matcher’s ability to segment glass-like objects. It identifies the feature directions corresponding to glass-like appearances by comparing features from glass or mirror regions using a manually labeled subset of 11 image pairs. The approach was evaluated on a subset of either 3 or 4 images from different datasets.

The reviews:
The main strengths are: 1）this paper addresses an interesting problem and the main weaknesses are: 1）Poor writing (bad structure, wrong references, missing definition of notations, grammar mistakes). 2）Insufficient experiments (anecdotal results, minor improvement, missing ablation study). All reviewers find this paper hard to read.

Due to insufficient contributions, the AC agrees with the reviewers and does not recommend accepting it at this conference.
The authors are encouraged to improve this work by making more substantial contributions to other venues.

**Additional Comments On Reviewer Discussion:**

The reviewers recognized that the main strengths are: 1）this paper addresses an interesting problem and the main weaknesses are: 1）Poor writing (bad structure, wrong references, missing definition of notations, grammar mistakes). 2）Insufficient experiments (anecdotal results, minor improvement, missing ablation study). All reviewers find this paper hard to read.

The authors did not provide feedback and the issues remain.

---

### Decision · Program_Chairs · 2025-01-22

Reject